# Behavior and Activity Patterns of the Critically Endangered Mangshan Pit Viper (*Protobothrops mangshanensis*) Determined Using Remote Monitoring

**DOI:** 10.3390/ani14152247

**Published:** 2024-08-02

**Authors:** Zeshuai Deng, Xiangyun Ding, Bing Zhang, Linhai Li, Dejia Hou, Yue Cao, Jun Chen, Daode Yang

**Affiliations:** 1Institute of Wildlife Conservation, Central South University of Forestry and Technology, Changsha 410004, China; 20210100016@csuft.edu.cn (Z.D.); 20221100047@csuft.edu.cn (X.D.); zhangbing@qlnu.edu.cn (B.Z.); 20211100052@csuft.edu.cn (D.H.); 20221200085@csuft.edu.cn (Y.C.); 2Department of Wildlife Conservation, State Forestry and Grassland Administration, Beijing 100714, China; 19313185719@189.cn; 3Administration Bureau of Hunan Mangshan National Nature Reserve, Chenzhou 423000, China; 13975590551@139.com

**Keywords:** snake activity rhythm, environmental factor, *Protobothrops mangshanensis*, Mangshan National Nature Reserve

## Abstract

**Simple Summary:**

Studying the behavior of animals is very important for their survival and protection. The Mangshan pit viper, a rare and endangered snake species from China, needs urgent conservation due to its very low population in the wild. To help with its conservation, we studied the behavior of 15 Mangshan pit vipers observed at different sites in Hunan Province during the summer and autumn of 2021. We looked at how environmental factors like temperature, humidity, and light affect the snakes’ activity during the day and night. We found that the snakes had specific behaviors like resting, sunbathing, moving, and others. The species exhibits three daily activity peaks, with the highest activity observed at 5:00–7:00, 9:00–11:00, and 18:00–20:00, which might be related to finding food and sunbathing. In addition, our research has found that the Mangshan pit viper is not a completely terrestrial or arboreal species, and in our study, this species is classified as semi-arboreal. We also found that humidity was a key factor influencing the snakes’ activity. There were small differences in behavior between the 15 snakes, but snakes from different habitats behaved differently. Our study helps us understand the Mangshan pit viper’s behavior better and provides information for developing effective conservation measures for this rare species.

**Abstract:**

This study focuses on understanding the behavior and activity patterns of the critically endangered *Protobothrops mangshanensis* in China in order to better provide scientific data for upcoming artificial breeding and propagation efforts. We conducted a long-term observation of 15 Mangshan pit vipers at different sites in Hunan Province during the summer and autumn of 2021. Our methods involved analyzing the influence of environmental factors such as temperature, relative humidity, and light condition on the snakes’ day and night activity and behaviors. The results revealed that the wild behaviors of *Protobothrops mangshanensis* include resting, sunbathing, crawling, and exploring, with distinct rhythms in their diel behavior. The snakes’ diel activity exhibits three peak periods which may be related to food activity and sunbathing. This study also highlights the complex interplay of environmental factors on the activity of *Protobothrops mangshanensis*. Relative humidity was identified as a critical factor accounting for the difference in activity between observation groups. There was little inter-individual variation among the 15 *Protobothrops mangshanensis*, even though these snakes used terrestrial and arboreal habitats under different environmental conditions. These findings enhance our understanding of *Protobothrops mangshanensis* behavior and provide a basis for effective conservation measures for this rare and critically endangered species.

## 1. Introduction

Animal behavior, the orchestrated and adaptive strategies employed by individuals and groups to secure survival and reproduction, serves as a cornerstone in comprehending species ecology and conservation [1,2,3,4]. By delving into animal behaviors, we gain invaluable insights into how organisms navigate their environments, adapt to varying conditions, and respond to external pressures [5,6,7]. Understanding species activity patterns is therefore paramount, especially for endangered species, as it informs the formulation of effective conservation strategies [8,9].

Among ectotherms, snakes exhibit a myriad of behavioral rhythms and patterns intricately linked to environmental factors [10]. Diel activity patterns, in particular, are of great interest as they illuminate a snake’s capacity to forage, mate, and evade predators [11,12]. However, despite their ecological importance, diel activity patterns remain elusive for many snake species, especially those on the brink of extinction [13].

The Mangshan pit viper (*Protobothrops mangshanensis*), a distinctive and critically endangered species endemic to China, exemplifies this knowledge gap [14]. Native to the Mangshan region, straddling Hunan and Guangdong provinces, this species confronts an array of existential threats. Listed as “globally endangered” on the IUCN Red List and designated a first-class protected animal in China [15], the wild population of *P. mangshanensis* hovers perilously low, estimated to comprise fewer than 500 individuals [16,17]. The primary threats to *P. mangshanensis* stem from human activities, notably habitat destruction and fragmentation driven by road construction, tourism expansion, and small hydroelectric power stations. These anthropogenic impacts have disrupted the species’ natural behavioral patterns and interactions, underscoring the urgency for a thorough understanding of its behavioral ecology. Previous research on *P. mangshanensis* has largely centered on venom properties, taxonomy, and habitat surveys, offering valuable yet limited insights. Notably absent from these studies is an in-depth examination of diel activity patterns, which are pivotal in comprehending the species’ survival strategies and the challenges it faces in the wild. Understanding these patterns is imperative for devising conservation measures that address the root causes of its endangerment [18].

In line with the requirements outlined in the “Fourteenth Five-Year Plan for Forestry and Grassland Conservation and Development of China”, it is imperative to implement rescue and protection efforts for 48 critically endangered wildlife species, including the giant panda (*Ailuropoda melanoleuca*), Asian elephant (*Elephas maximus*), Hainan gibbon (*Nomascus hainanus*), Northeast tiger (*Panthera tigris* ssp. *altaica*), Chinese pangolin (*Manis pentadactyla*), four-toed terrapin (*Testudo horsfieldii*), and the Mangshan pit viper (*Protobothrops mangshanensis*), among others. Key habitats for these species must be delineated and strictly protected, ecological corridors connected, and an increase of 10% in the area of these critical habitats ensured. To provide long-term survival guarantees, breeding bases for rare and endangered wild animals and genetic resource banks should be established. Furthermore, reintroduction efforts should be carried out for 15 rare and endangered wild animals such as the giant panda, Przewalski’s horse (*Equus ferus przewalskii*), Pere David’s deer (*Elaphurus davidianus*), and crested ibis (*Nipponia nippon*), helping them reacclimate to the natural environment and increasing their wild population numbers. For species like the Mangshan pit viper that are not yet mature for breeding or reintroduction, scientific methods for their artificial breeding and reintroduction should be explored (https://www.gov.cn/xinwen/2021-08/19/content_5632156.htm, accessed on 5 July 2024).

Our study aims to bridge the knowledge gap by exploring the factors that significantly influence the behavior of *P. mangshanensis* under natural conditions and determining the optimal range for these factors. To achieve this, we confronted the unique challenges posed by the species’ endangered status and stringent management measures. Instead of the conventional approach of subcutaneously implanting radio transmitters, which was deemed infeasible due to the species’ sensitivity and regulatory constraints [10], our study innovatively adapted techniques from captive animal studies and conservation monitoring. This approach integrates remote monitoring cameras for non-invasive real-time observation [19,20] and instantaneous scan sampling methods to systematically record behavior [21,22,23] at specified intervals. Though unconventional for snake behavior studies [17,18], it proved effective in mitigating logistical and ethical hurdles [19,20,21,22,23]. It empowered us to gain profound insights into the diel rhythms, arboreality, and behavioral adaptations of *P. mangshanensis*, thereby guiding the design of artificial breeding landscapes and optimal reproductive conditions tailored to both terrestrial and arboreal habitats. Ultimately, our findings contribute to the informed development of conservation strategies aimed at restoring and safeguarding this unique and imperiled species.

## 2. Materials and Methods

### 2.1. Study Area 

This study was conducted in the Mangshan National Nature Reserve of Hunan Province, China, located between 112°43′19″ E and 113°0′10″ E, and between 24°52′0″ N and 25°23′12″ N. The Mangshan National Nature Reserve boasts diverse forest types, including lowland evergreen broadleaf forests, mid-mountain mixed evergreen and deciduous broadleaf forests, and mid-mountain coniferous–broadleaf mixed forests. It is located in a subtropical monsoon climate zone, with an annual average temperature of approximately 17.2 °C and annual precipitation exceeding 1700 mm (in the past 10 years), fostering exceptional biodiversity [24]. Geographic coordinates were recorded using the WGS84 datum, as previously described by Zhang et al. (2020) [18]. The study focused on 15 sites within the reserve, ranging in elevation from approximately 700 to 1200 m and encompassing diverse vegetation types such as evergreen broad-leaved forests, shrub bamboo forests, and mixed coniferous and broad-leaved forests (Figure 1). Due to the restricted home ranges and behavioral habits of the Mangshan pit viper, coupled with individual identification through unique head markings, the 15 snakes observed at these 15 locations were confirmed to be 15 distinct individuals [18]. 

### 2.2. Observation Method

Given the rarity and strict protection status of adult Mangshan pit vipers, radio-tracking methods were deemed inappropriate [21]. Therefore, we resorted to a combination of instantaneous scan sampling and all-occurrence observation methods, with necessary modifications to minimize human interference. This method involves humans recording all displacements along the species’ travel route at regular intervals (30 min), as well as documenting all behavior patterns and arboreality patterns of the animals during each time period [22,23].

Upon receiving reports regarding the discovery of adult Mangshan pit vipers from the reserve’s forest patrol team, which conducts a comprehensive survey of the entire reserve every two weeks, research personnel promptly set up cameras near the paths usually followed by the snakes so that they could film them.

To conduct observations, we installed 2–3 infrared monitors (Xiaomi Corporation^TM^, Shenzhen, China, C700) at a distance of 3–5 m from each snake. These monitors were accompanied by remote temperature, humidity, and light intensity recorders (Prsens^TM^, Jinan, China, SN-3002-GZ-RS485) placed at the same location. Researchers monitored the snakes in real time from a distance of approximately 1 km away, utilizing the infrared monitors and recorders to document their behavior patterns. The environmental factors that have been shown to significantly influence snake activity include temperature, relative humidity, prey availability, and light intensity [25]. Consequently, we collected data on season, location (locus), per hour temperature, per hour relative humidity, per hour total illumination, hour of the day, and snake ID as analytical factors to investigate their impacts on the behavior of Mangshan pit vipers.

Every 30 min, researchers recorded the temperature, humidity, and light intensity from the remote recorders. To cover the entire motion path of the Mangshan pit vipers observed during the surveillance period as much as possible, they approached the snakes to within approximately 5 m, using an infrared rangefinder (ORPHA^TM^, Munich, Germany, DB550L+) to manually measure the displacement of the snakes. When the snakes moved out of the range of the infrared monitors, additional monitors were set up at a distance of 3–5 m from their new location. If the snakes entered caves, rocky outcrops, or areas where continued monitoring was not feasible (e.g., across cliffs), monitoring continued at the last known location for 24 h. If the snakes reappeared within this period, monitoring was resumed; otherwise, it was discontinued. It should be noted that because snakes are ectotherm animals, infrared surveillance cameras are used for monitoring them in low light conditions at night. However, infrared surveillance is ineffective for most observations during the period from one hour after sunset to one hour before sunrise (roughly around 8:30 p.m. to 5:00 a.m. local time each day). In this study, we define a state as stationary and without movement if the observed target’s position and posture remain unchanged from the moment it becomes completely invisible due to low light conditions until the next time it becomes visible again.

We define the behavior of a snake coiling its body on a branch more than 2 m above the ground as arboreal, while the behavior of its body being entirely on the ground is defined as terrestrial [26].

### 2.3. Data Exclusion

All observations with continuous tracking times less than 24 h were excluded from the analysis, ensuring that only data from observations exceeding 24 h were utilized. Given that the official transition from summer to autumn, as designated by the National Meteorological Center, falls on August 7th at 14:53:48 with the start of autumn, it is difficult to categorize the data collected on that day as belonging solely to summer or autumn, since it encompasses less than 24 h of either season. Therefore, the data collected on that day were not included in our analysis either. From June 25th to September 3rd, observations were conducted on a total of 70 days. However, after careful analysis, only 37 days were retained as complete observations of the snakes.

### 2.4. Data Analysis

In this study, the movement frequency of the Mangshan pit viper was calculated by dividing the recorded movement time by the total recording time. During the recording process, it was observed that the snake’s activity frequency at night may have overlapped with the active period of its primary prey, *Leopoldamys edwardsi*, a rodent species [13]. To verify this conjecture, we obtained activity frequency data for *Leopoldamys edwardsi* during the summer and autumn seasons of 2021 from Professor Wang Bin’s team at Hunan Normal University, who had also collected rodent data in the same area. They calculated the movement frequencies based on the number of movement photos captured by triggered infrared cameras within a specific time frame (1 h), divided by the total number of photos taken of the species during that period [24]. To analyze the overlap in movement frequency between the Mangshan pit viper and *Leopoldamys edwardsi* during the summer and autumn seasons from 18:00 to 20:00, we employed the Kernel Density Estimation (KDE) method. Using the ‘overlap’ package in R, we calculated the overlap coefficient with a 95% confidence interval (95% CI) based on 9999 bootstrap resampling calculations of the Delta4 value. The overlap was defined as “high” when Delta4 > 0.9, “moderate” when 0.9 > Delta4 > 0.8, and “low” when Delta4 < 0.8 [27]. The frequency of movement was described using mean ± standard deviation.

In order to determine the influence of environmental factors on the habitat preference (arboreal or terrestrial) of the snakes, we used analysis generalized linear models (GLMs) from the lme4 package in R [28]. These models examined the relationship between habitat preference and various factors such as season, location (locus), per hour temperature, per hour relative humidity, per hour total illumination, hour of the day, and snake ID (treated as a random effect). Given the correlation among factors like hourly temperature, hourly relative humidity, luminosity, season, and time of day, as well as their nested nature, two model approaches were adopted:A main effects model: Arboreal or terrestrial (binary outcome)~season + per hour temperature + per hour relative humidity + per hour total illumination + hour of the day (1 to 24) + (1|snake ID).An interaction effects model: Arboreal or terrestrial (binary outcome)~season × per hour temperature × per hour relative humidity × per hour total illumination ×hour of the day (1 to 24) + (1|snake ID).

Both models were compared based on their residual sum of squares to determine the most appropriate model for further analysis [25,29].

A similar analysis was used to determine the influence of environmental factors on movement occurrence, behavior patterns, and movement distance. For movement occurrence and behavior patterns, binomial distributions were specified for the family parameter, considering movement patterns. For movement distance, which did not follow a normal distribution, a Poisson distribution was chosen for the family parameter, with habitat type included as an additional predictor. The analysis of movement types also utilized a Poisson distribution due to its non-normal distribution. In addition, to determine the effects of environmental conditions on arboreality by snakes, we used glmmTMB in R and applied Multinomial Logistic Regression within a GLMM framework to study habitat type selection by snakes, specifying the family as categorical. The model considered fixed effects like season, location, environmental variables (temperature, humidity, and illumination), and time of day, along with snake ID as a random effect. Furthermore, we acknowledged the potential for interactions among these variables and, although we may have ultimately included only those interactions that were statistically significant or theoretically important to avoid overparameterization, we considered their inclusion in the initial model specification [30]. In these generalized linear models, if the *p*-value < 0.05, it indicates that this factor has a significant impact on the arboreality, movement, or behavior patterns of the species. The interaction model considers the mutual influences between factors. If the direct model outperforms the interaction model, it suggests that the interactions between factors do not significantly affect the final arboreal habit of the snakes in this study. Under the conditions of this study, it is considered that the arboreality, movement, or behavior patterns of the snakes are influenced by the factors with significant results.

## 3. Results

### 3.1. Habitat Preference and Diel Activity Patterns

*Protobothrops mangshanensis* specimens measured 143 ± 26 cm in length, indicating that they were all adult snakes. However, the sex of the snakes remained undetermined during the observation period, which ranged from 3 to 6 days. The profound influence of specific factors within arboreal (Figure 2A,B) and terrestrial (Figure 2C,D) habitats on the snakes’ diel activity patterns was observed.

Mangshan pit vipers allocated their time between the arboreal and terrestrial habitats in approximately equal proportions, occupying arboreal habitats for 54.91% and terrestrial habitats for 45.09% of the total observation period. During the summer, our 642 h of observations revealed a clear preference for arboreal habitats, with 56.00% (*N* = 359.5 h out of a possible 642 h) spent in trees. Notably, 91.09% (*N* = 327.5 h/359.5 h) of these arboreal activities occurred during the peak hours of 12:00–15:00 p.m., suggesting a midday preference for this habitat. Conversely, terrestrial perching was observed for 565 h, accounting for 44.00% of the summer observations, with the majority (66.01%, *N* = 186.5 h/282.5 h) occurring at night and a smaller proportion (20.00%, *N* = 56.5 h /282.5 h) during the early morning hours of 9:00–11:00 a.m.

In autumn, we accumulated 291.5 h of observations, during which the Mangshan pit vipers were detected 81.82% (n = 238.5 h/291.5 h) of the time. Similar to summer, arboreal activity persisted, accounting for 51.99% (142 h out of 238.4 total hours), with a pronounced midday peak from 12:00 to 15:00 p.m. (91.93%, n = 114 h/142 h). Terrestrial perching, though less frequent than in arboreal habitats, still constituted 48.00% (n = 114.5 h/238.5 h) of autumn observations, with a notable shift towards nocturnal activity (73.79%, n = 84.5 h/114.5 h) and a smaller diel peak at 9:00–11:00 a.m. (26.20%, n = 30 h/114.5 h).

Despite seasonal variations, the GLM indicated that season did not significantly influence the Mangshan pit vipers’ habitat preference (Table 1, *p* = 0.2893 > 0.05). Instead, humidity and light emerged as the primary factors governing their activity patterns.

### 3.2. Diel Activity Rhythms and Movement Peaks

The diel activity rhythm of Mangshan pit vipers exhibited remarkable consistency, with resting (accounting for 83.45 ± 0.83% of their daytime behavior) and basking (comprising 7.31 ± 0.86%) predominating around the 9:00 a.m. to 11:00 a.m. timeframe. Our observations over 24 h periods revealed three distinct peaks of movement activity (3.78 ± 0.58%), occurring at 5:00 a.m.–7:00 a.m., 9:00 a.m.–11:00 a.m., and 17:00–19:00 (Figure 3). Notably, all individuals followed this consistent diel rhythm, displaying no significant variations in their activity patterns.

According to the overlap index calculations, the degree of overlap in activity time between the Mangshan pit viper and *Leopoldamys edwardsi* from 18:00 to 20:00 during both the summer and autumn seasons was moderate, with values of 0.85 (95% CI: 0.79–0.91) in summer and 0.83 (95% CI: 0.76–0.90) in autumn. This suggests a possible correlation between the snakes’ hunting behavior and the availability of their prey. Furthermore, the middle peak of movement activity coincided with the snakes’ basking periods (Figure 4), indicating a balanced schedule of foraging and thermoregulation.

The statistical analysis using fitting models revealed that both the occurrence and distances of movement were influenced by temperature and the time of day. Additionally, seasonal variations were found to significantly impact movement distances (Table 2 and Table 3). Similarity, Model 2, which accounted for interaction effects among variables, demonstrated a lower Rss compared to Model 1, which only considered main effects. However, both models were deemed statistically equivalent (*p* = 0.631 > 0.05), highlighting the complex interplay of environmental factors influencing the Mangshan pit vipers’ diel activity rhythms and movement patterns.

### 3.3. Environmental Influences on Behavior

Light intensity, the time of day, and seasonality were key influencers (Table 4). We chose the general generalized linear model because its RSS was lower than the interactive linear model (*p* = 0.187 > 0.05). In all linear relationships examined, none of the individual ID demonstrated a significant impact on the analysis target (Table 1, Table 2, Table 3 and Table 4).

About the temperature effects on activity levels, we observed peak activity within an optimal temperature range of 22–26 °C, beyond which the snakes became inactive. Specifically, no movement was recorded when temperatures dipped below 16°C or soared above 31 °C (Figure 5). Notably, as temperatures escalated beyond 29–33 °C, the snakes abandoned their basking pursuits and sought refuge in the shade, highlighting their sensitivity to extreme heat. The duration of basking sessions was inversely proportional to the magnitude of temperature fluctuations, with the longest basking durations observed at moderate temperatures of around 14 °C, lasting approximately 5 h from 9:00 a.m. to 3:00 p.m. Conversely, basking became increasingly rare at temperatures exceeding 30 °C (Figure 6), further emphasizing the intricate balance between the snakes’ thermoregulatory needs and the challenges posed by varying environmental conditions.

Furthermore, our observations during autumn reveal a subtle yet discernible shift in the snakes’ basking behavior. Mangshan pit vipers spent marginally more time basking in the warm autumnal sun compared to the summer months (Figure 7). This heightened propensity for basking, statistically significant at *p* < 0.05, underscores the strategic adaptation of these snakes to seasonal variations. In stark contrast to summer, when the snakes exhibited a diminished tendency to bask during peak sunlight hours (10:00 to 14:00), autumn witnessed a pronounced inclination towards sunny locales for thermoregulation (Figure 8). This seasonal shift highlights the snakes’ exquisite ability to optimize their basking behavior in response to changing environmental conditions.

Throughout the 37 d study period, the environmental conditions in the study area remained relatively stable yet dynamic, providing valuable insights into the adaptability of Mangshan pit vipers. The diel mean air temperature hovered around a comfortable 22.52 ± 1.52 °C (mean ± standard deviation), indicating a moderate climate conducive to the snakes’ activity. During our research period, the recorded average daily temperature in summer was 22.3 ± 5.4 °C, and the average daily temperature in autumn was 18.8 ± 7.2 °C. In parallel, the relative humidity maintained a high and stable level of 92.28% ± 5.97%, reflecting the humid environment characteristic of the region. The illumination levels, measured at 26,296.17 ± 8263.67 lux (mean ± standard deviation), fluctuated throughout the day, offering varying degrees of sunlight exposure for the snakes to bask in.

Despite the occasional interruption of rainfall, which was recorded on 18 d out of the total study period, the snakes were observed in motion on 32 d, accounting for 68% of the study duration, showcasing their active lifestyle. The snakes covered an average diel distance of 16.15 ± 14.78 m (2.50 m–55.56 m), underscoring their mobility and the wide range of their movements within their habitat. This variability in diel distances traveled highlights the snakes’ ability to adjust their foraging and exploration patterns in response to changing environmental conditions and resource availability.

## 4. Discussion

Our study on the critically endangered *P. mangshanensis* revealed several key insights into its behavior, activity, and habitat selection patterns. Delving deeper into the model statistics, it becomes evident that a myriad of environmental factors intricately shapes the behavior of Mangshan pit vipers. Notably, the snake’s arboreality mode also exerts a significant impact on its activity patterns, while temperature and humidity, though influential, assume a more subordinate role.

Mangshan pit vipers possess a relatively robust body, adapted for crawling and predation on the ground. Their body length can often reach up to 2 m, with a relatively broad and triangular-shaped head. The proportion of their body length to diameter is moderately balanced, and their scales are typically hard and closely arranged. Originally, it was anticipated that this species of snake primarily leads a terrestrial lifestyle [13]. Notably, we observed that these snakes did not exhibit a clear preference for either arboreal or terrestrial habitats, differing from the expected trend based on their body morphology. Temperature emerged as the primary factor influencing their behavior, with humidity and light intensity also playing significant roles. Furthermore, we identified distinct diel activity rhythms, characterized by peaks in crawling, basking, and exploratory behaviors, with the latter correlated with prey availability.

One of the intriguing findings of our study was the habitat selection of Mangshan pit vipers, which, despite not fitting the typical morphological characteristics of slender-bodied arboreal snakes, exhibited a higher degree of arboreal activity compared to many strictly terrestrial species [31,32,33,34]. This led us to suggest categorizing *P. mangshanensis* as semi-arboreal, acknowledging their unique behavioral tendencies that do not neatly fit into either strictly arboreal or terrestrial categories. Furthermore, their sedentary nature and prolonged resting periods during diel activities align closely with those of terrestrial snakes such as keelback snakes (*Hoplocephalus bungaroides*) and *Crotalus horridus* [35,36,37]. However, our observations revealed that Mangshan pit vipers spend even less time actively moving than these latter species, emphasizing the uniqueness of their behavioral patterns. Thus, while their resting behaviors mirror those of terrestrial snakes, their arboreality and activity levels suggest a more nuanced classification that acknowledges both arboreal and terrestrial tendencies.

Consistent with previous research, our analysis identified temperature as the most influential factor shaping the behavior of Mangshan pit vipers [38]. By basking in warm areas and seeking shelter in cooler ones, these snakes actively regulate their body temperatures, which in turn affects their overall activity levels, and temperature drops in autumn so snakes spend more time basking to warm up [39]. This finding is in line with our previous observations using the Maxent model, which revealed that the distribution of Mangshan pit vipers is notably influenced by two key climate variables: precipitation during the driest month (typically occurring in winter) and the maximum temperature of the warmest month (representative of summer temperatures) [12]. Given that our study was conducted during summer and early autumn, it is consistent with our previous research as well as findings from other studies where temperature emerged as a prominent factor [12,40]. We speculate that a temperature range of 23–27 °C is optimal for *P. mangshanensis*, as they exhibited increased activity and reduced basking at temperatures above and below this range. It is worth noting that excessively high temperatures can potentially have adverse effects on snakes [40].

While temperature was the primary driver, humidity and light intensity also played significant roles in shaping the behavior of Mangshan pit vipers. High humidity, often associated with increased precipitation, has been shown to correlate with snake activity [41], potentially due to the need for snakes to seek drier microhabitats [42]. Some studies suggest that denser tree canopies result in less light reaching the forest floor, and that the amount of light on the canopy is higher than that on the ground [43,44]. Similarly, light intensity influenced habitat preference, with tree canopies providing superior basking opportunities [45]. This importance of sunlight for thermoregulation and overall fitness underscores the need to consider light availability in habitat restoration efforts [46]. This may potentially explain why Mangshan pit vipers exhibit semi-arboreal behavior in the Mangshan area, given its high forest coverage (>97%) [47], However, this is just a speculation, and further research is needed to confirm this hypothesis.

Our findings revealed distinct diel activity rhythms in Mangshan pit vipers, characterized by three primary peaks. Crawling activity peaked during early morning and late afternoon, with a basking peak occurring in the morning. These rhythms reflect the snakes’ adaptation to the surrounding environment and the influence of both biotic and abiotic factors [48]. Notably, an additional peak in exploratory behavior was observed in the afternoon during these months, aligning with the activity patterns of local rodents and suggesting a link to prey availability [49]. Intriguingly, these movement peaks aligned with the activity patterns of *Leopoldamys edwardsi*, a rodent species that is the most common in our study area [24] and serves as a primary prey species for the Mangshan pit viper, according to previous studies which have shown that the viper’s diet is dominated by small mammals, with occasional predation on birds [13]. The alignment suggests a possible correlation between the snakes’ hunting behavior and the availability of their prey, which is particularly intriguing because it could indicate that if snakes adjust their activity to mirror that of their prey, they are employing a highly adaptive hunting strategy [50,51]. And species themselves exhibit different behaviors to adapt to environmental factors, such as seeking suitable temperatures, searching for water sources, and avoiding humidity. Therefore, the behaviors of these species reflect their adaptability and tolerance to the similar environments [52,53].

Lastly, it is important to acknowledge the limitations of our study. One significant limitation is the lack of gender identification for the individuals included in our research. As sexual dimorphism [54,55,56], or differences in physical characteristics between males and females, is known to exist in some snake species, it is possible that gender-specific behavioral patterns or habitat preferences may have gone undetected in our study. Many studies have shown that the sexes of snakes have an impact on various aspects of their behavior. For example, the shy-social behavior of *Thamnophis sirtalis* is influenced by gender [57]. *Aipysurus laevis and* Sword Snake exhibit obvious sexual dimorphism in morphology, and the behavioral differences between their gender and body size have a positive effect on locomotor ability and metabolic rate. Therefore, males of the same age tend to prefer exercise [58,59]. A gender-based difference also affects vulnerability to tides in *Hydrophiinae, Elapidae* [60]. However, due to the constraints of our research conditions, we were unable to conduct gender identification for the Mangshan pit vipers under investigation. This lack of gender data may limit the scope and generalizability of our findings and highlights an area for future research to explore more thoroughly. Furthermore, the relatively short observation periods of 3 to 6 days per snake, combined with the absence of gender-based grouping, may have contributed to the lack of significant linear relationships between individual snakes (identified by their IDs) and their movement or behavioral patterns. The limited sample size and observation duration, while sufficient for our primary objectives, may not have been adequate to detect subtle gender-specific or individual-level variations that could have emerged over a longer study period or with a larger cohort.

## 5. Conclusions

In conclusion, the establishment of an artificial breeding base for Mangshan pit vipers necessitates a comprehensive approach that addresses both their arboreal and terrestrial habitat preferences. To ensure the snakes’ arboreal needs are met, an abundance of trees should be incorporated into the breeding environment. Additionally, to cater to their terrestrial requirements, the base should provide caves, shelters, and other suitable landscaping features. Furthermore, maintaining a temperature range of 24~27 °C is crucial for maintaining the snakes’ activity levels and overall well-being.

While our study did not provide definitive guidance on humidity levels due to a lack of significant findings on its impact outside of hibernation periods, it is essential to strive for a relatively stable humidity to support the snakes’ physiological functions. Given the species’ distinct diel activity rhythms, it is recommended to provide ample sunlight during their basking peak hours, specifically from 11 a.m. to approximately 1 p.m., to support their thermoregulatory needs.

Moreover, considering the snakes’ exploratory and potentially prey-related activity peaks in the afternoon, food should be offered during a time that overlaps with their natural foraging rhythms, such as between 7 p.m. and approximately 9 p.m. By incorporating these environmental and feeding strategies into the design and management of the breeding base, we can better support the conservation efforts for this critically endangered species and enhance their chances of survival and reproduction in captivity.

## Figures and Tables

**Figure 1 animals-14-02247-f001:**
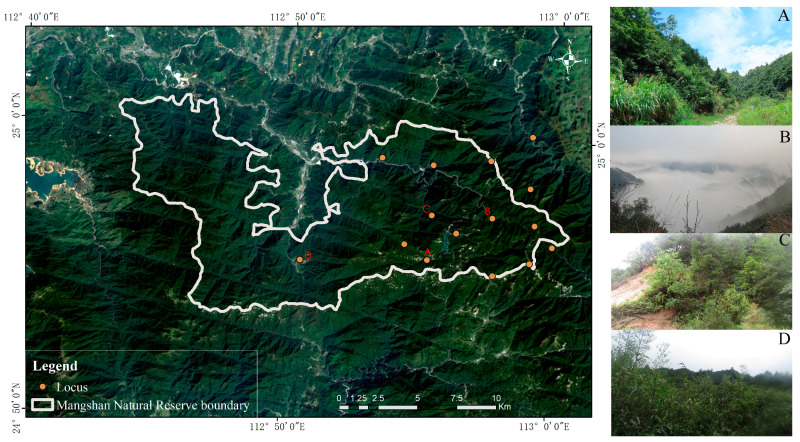
The research sites for the wild Mangshan pit vipers encompassed diverse habitats, each captured in a series of photographs by Bing Zhang labeled (**A**–**D**). These show the surrounding habitats where the Mangshan pit vipers were discovered, and the locations are marked on the map.

**Figure 2 animals-14-02247-f002:**
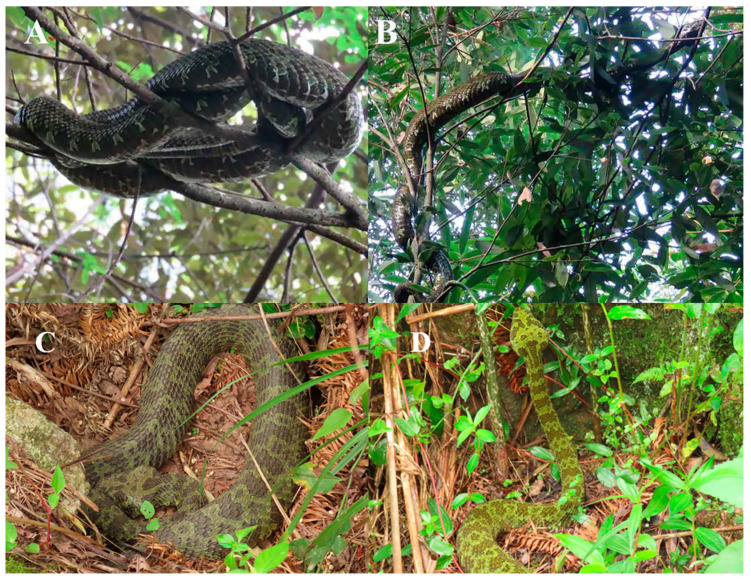
Behaviors of Mangshan pit vipers in their natural habitat; photographs by Bing Zhang. (**A**) Arboreal resting, (**B**) arboreal locomotion, (**C**) terrestrial resting, and (**D**) terrestrial locomotion.

**Figure 3 animals-14-02247-f003:**
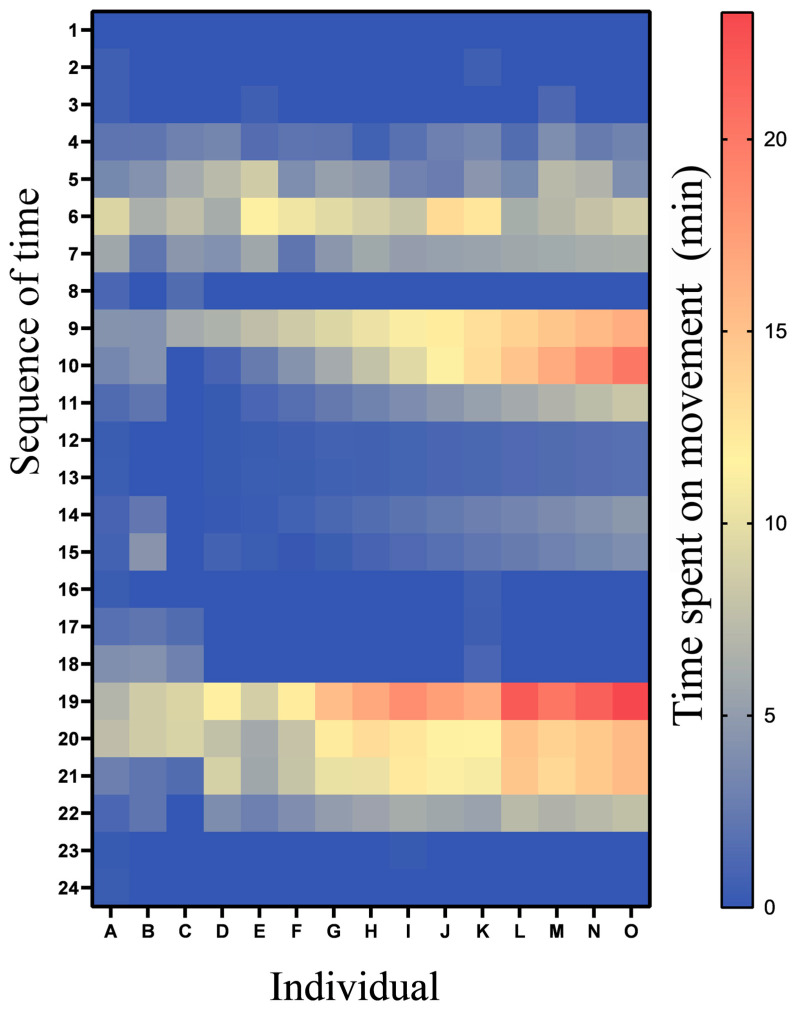
Comparative examination of individual Mangshan pit viper diel activity time (mean) in natural habitats. A~O: 15 individual Mangshan pit vipers (SD in Appendix A).

**Figure 4 animals-14-02247-f004:**
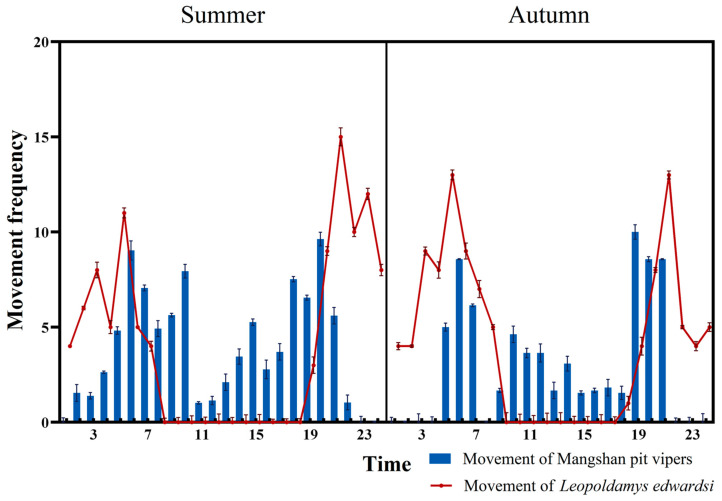
Concordance in movement peaks of Mangshan pit vipers with known activity patterns of rat prey (Mean ± SD).

**Figure 5 animals-14-02247-f005:**
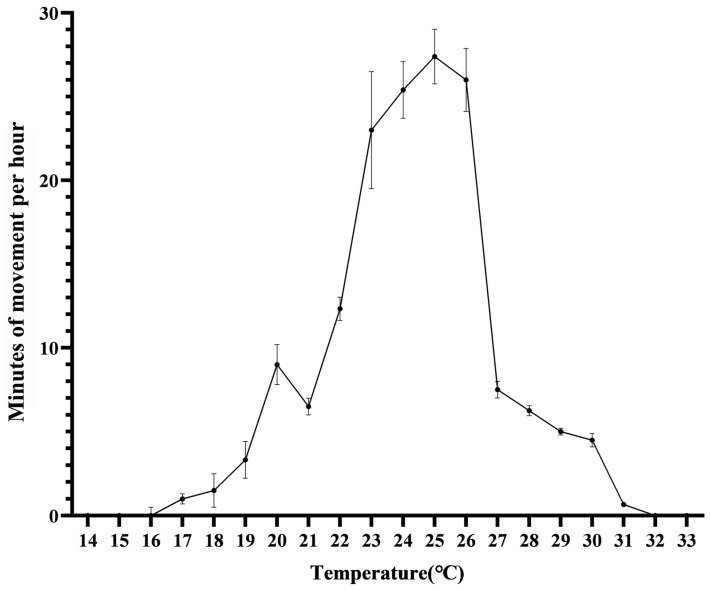
Relationship between ambient temperature and hourly movement counts of Mangshan pit vipers (mean ± SD).

**Figure 6 animals-14-02247-f006:**
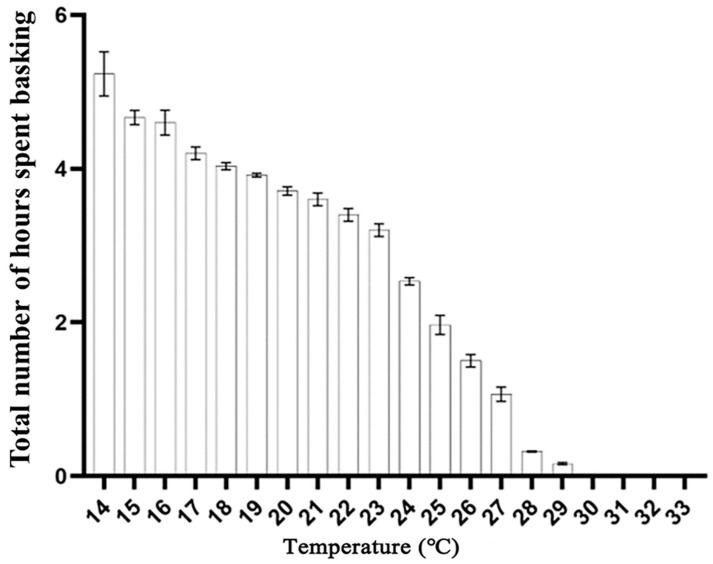
Basking behavior for Mangshan pit viper across varying temperature ranges (14~30 °C) in the wild (mean ± SD).

**Figure 7 animals-14-02247-f007:**
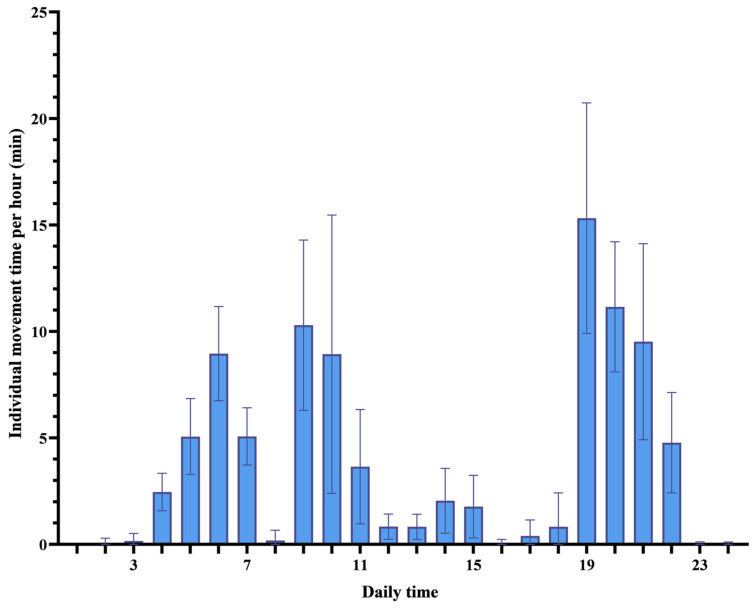
Consolidated view of diel behavioral patterns, highlighting collective pit viper strategies (mean ± SD).

**Figure 8 animals-14-02247-f008:**
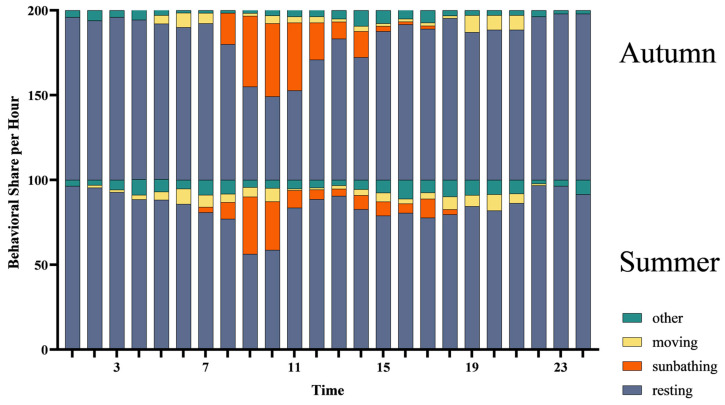
Behavioral frequency distribution (mean), encompassing a broad spectrum of activities in natural settings of summer and autumn (SD in Appendix A).

**Table 1 animals-14-02247-t001:** Generalized linear model (GLM) based on the impact of various environmental factors on the arboreality of Mangshan pit vipers across two distinct seasons.

Parameter	Coefficient	Standard Error	Lower 95% CI	Upper 95% CI	*p*
(Intercept)	−189.139	18,462.70	−1669.30	2047.58	0.8334
ID	30.129	10.236	18.332	67.569	0.8724
Per hour temperature	141.265	38.68	67.91	214.62	0.0003 ***
Per hour relative humidity	35.613	15.84	5.58	65.64	0.1158
Per hour total illumination	−76.31	47.24	−165.89	13.27	0.0245 **
Hour of the day	−2.977	2.81	−8.30	2.35	0.1063
Season ^a^	−64.178	104.70	−262.72	134.36	0.2893

a: summer = 1; autumn = 0. *** *p* < 0.001; ** *p* < 0.01. CI: Confidence interval.

**Table 2 animals-14-02247-t002:** Generalized linear model (GLM) based on arboreality preferences and environmental factors influencing whether Mangshan pit vipers move in two seasons.

Parameter	Coefficient	Standard Error	Lower 95% CI	Upper 95% CI	*p*
(Intercept)	1.374	1.623	−1.714	4.461	0.397
ID	9.676	3.145	−1.442	13.279	0.443
Per hour temperature	−0.014	0.007	−0.028	0.010	0.046 *
Per hour relative humidity	−0.028	0.043	−0.109	0.052	0.503
Per hour total illumination	−0.004	0.010	−0.023	0.015	0.694
Hour of the day	0.380	0.138	0.117	0.642	0.006 **
Season ^a^	0.150	0.129	−0.094	0.395	0.243

a: summer = 1; autumn = 0. ** *p* < 0.01; * *p* < 0.05. CI: Confidence interval.

**Table 3 animals-14-02247-t003:** Generalized linear model (GLM) based on the impact of various environmental factors on habitat preference (arboreal versus terrestrial) of Mangshan pit vipers across two distinct seasons.

Parameter	Coefficient	Standard Error	Lower 95% CI	Upper 95% CI	*p*
(Intercept)	−98.116	42.244	−205.669	22.108	0.200
ID	32.221	18.675	12.341	42.213	0.560
Per hour temperature	3.443	3.638	1.268	5.516	0.002 **
Per hour relative humidity	−0.045	1.099	−0.473	0.513	0.862
Per hour total illumination	−0.264	0.260	−0.609	0.117	0.163
Hour of the day	16.097	0.189	8.335	22.108	0.000 ***
Season ^a^	17.072	3.336	10.224	23.071	0.009 **

a: summer = 1; autumn = 0. *** *p* < 0.001; ** *p* < 0.01. CI: Confidence interval.

**Table 4 animals-14-02247-t004:** Generalized linear model (GLM) based on impact of various environmental factors on habitat preference (arboreal versus terrestrial) of Mangshan pit vipers across two distinct seasons.

Parameter	Coefficient	Standard Error	Lower 95% CI	Upper 95% CI	*p*
(Intercept)	−0.840	1.002	−2.447	0.768	0.317
ID	2.363	2.155	−1.826	10.441	0.423
Per hour temperature	−0.029	0.016	−0.060	0.003	0.039 *
Per hour relative humidity	0.017	0.006	0.006	0.028	0.043 *
Per hour total illumination	0.081	0.018	0.047	0.115	0.000 ***
Hour of the day	0.004	0.001	0.002	0.006	0.000 ***
Season ^a^	−0.180	0.036	−0.250	−0.111	0.000 ***
Arboreal or terrestrial ^b^	0.003	0.000	0.000	0.000	0.007 **

a: summer = 1; autumn = 0. b: terrestrial = 1; arboreal = 0. *** *p* < 0.001; ** *p* < 0.01; * *p* < 0.05. CI: Confidence interval.

## Data Availability

The data presented in this study are available on request from the corresponding author due to confidentiality agreements signed with the Hunan Provincial Forestry Bureau.

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
