# Peer review of "Behavior and Activity Patterns of the Critically Endangered Mangshan Pit Viper (Protobothrops mangshanensis) Determined Using Remote Monitoring"

_animals, 2024, doi:10.3390/ani14152247_

Round 1
Reviewer 1 Report
Comments and Suggestions for Authors
This paper describes some aspects of the natural history of a critically endangered and little-known snake species. The study is based on limited data from a small number of animals, but that is understandable given the limitations of interacting with P. mangshanensis, let alone the difficulty of locating them in the first place. The authors have done an impressive job of working around those limitations… if anything, their remote-sensing approach is notable enough that I might work it right into the title – “Activity patterns of the critically endangered Mangshan pit viper determined using remote monitoring” or something similar.
The methodology used here is all sound, and the paper is very well-organized and written. I do have some suggestions for the authors to consider, presented chronologically below… most should be fairly minor to address. My only over-arching concern is that the authors seem to focus a heavily on their statistical results, while most readers will probably be more interested in the BIOLOGICAL patterns they are testing. I strongly suggest reorganizing the Methods and Results to emphasize what each analysis is testing, and what the results mean in light of Mangshan pit vipers. Statistics are the tool, not the goal themselves.
This work represents a valuable contribution to our understanding (and hopefully the conservation) of a critically endangered species. I encourage the authors to just consider some adjustments for clarity and readability, with the goal of producing a stronger manuscript.
Line 24 – Change “Focuses on…” to “This study focuses on…”
35 – I would probably rephrase this a bit. It’s not that snakes from terrestrial and arboreal habitats used different activity patterns, it seems like snakes used terrestrial and arboreal habitats under different environmental conditions.
36 and throughout – Really, I usually see the term “habitat usage” used when quantifying the actual macrohabitat or microhabitat… substrate, structural features like rocks and logs, canopy coverage, etc. I think the authors are clear enough about what they’re doing here that the term is fine, but I would probably just change it to “arboreality” to avoid any ambiguity.
This also makes me wonder if the authors don’t have enough data for a subsequent study of microhabitat usage by P. mangshanensis. There’s probably an awful lot of microhabitat data in their snake videos.
50 and throughout – The word “diurnal” means “during the daytime” (as opposed to nocturnal, at night). I assume here the authors are talking about an animal’s 24-hour cycle, so “diel” is the appropriate technical term. Something less formal like “temporal” or “daily” would also be fine.
54 – Italicize Protobothrops mangshanensis.
71-75 – You should briefly explain what that Five-Year Plan is and what its goals are. Most international readers will be unfamiliar with it and will not be able to read the linked website.
89 – Again diel and not diurnal; you’re also (as above) not really quantifying habitat utilization. I would specify it as just “arboreality” or something similar.
96-106 – Your study site may be well described in Zhang 2020, but you should explain a little more about it here. There are several forest types, but is it generally dry forest, temperate forest, rainforest? At least there should be temperature and precipitation range. You want your readers to have a general sense of the area, as very very few will be previously familiar with the Reserve.
116 – I am not sure what “instantaneous scan sampling” and “all occurrence observation methods” are. You should link these terms to your actual methods, though I’m not sure that they’re entirely helpful or necessary at all.
126 – Using infrared cameras is interesting. Do snakes show up well using thermal imaging, being ectotherms? Or were they used just to allow monitoring in low-light conditions at night?
135 – From the way it’s used later, “displacement” needs some clarification here. When you are measuring displacement… are you quantifying ALL movements over the 24 hour period, or straight-line displacement from the original location? For example… if a snake moves 5m to the east, and stops; then later it moves 2m to the west (back towards its original starting point), and stops… that is a total of 7m movement, correct? Not a linear displacement of 3m from the original starting point. This is fine, but should be clarified. A very simple explanation is fine, but “document their displacement” is too vague.
147 – I cannot understand why August 7th data would be excluded. Why not just assign that day to Autum?
151 – Frequency of movement is different from displacement (distance of movement), and not mentioned in the Methods. Are you talking about the number of movements, or whether the snake moved at all? You really need to be very careful, specific, and consistent with the terminology of your methodology.
159-164 – This is a good example of what I mentioned earlier about emphasizing statistics vs. their biological meaning. These models are well-designed, and stating their parameters is probably useful to those interested in similar methods. But what do they MEAN biologically? What does a main-effects vs. an interaction-effects model imply is important to a snake’s activity? I urge the authors to carefully consider that many readers will likely be more familiar with and more interested in animal-natural-history aspects, so think about presenting your methods and results with that perspective in mind. “In order to determine x, we used analysis y.”
168-180 – Same comment as above. Ex. On line 168… to do WHAT with movement and habitat data? What exactly will these analyses show? Ex. line 174… for example, I might say something like “To determine the effects of environmental conditions on arboreality by snakes, we used Multinomial Logistic Regression within a GLM framework… etc.” This is much better than saying “we used this approach for these data” Instead it helps orient readers to understand exactly what is being tested, what that analysis does, and what they can expect the results to mean.
I again urge the authors to consider this for all their analyses. Remember, NOBODY understands this stuff as well as you do. When you’re so close to the study for so long, it’s easy to lose sight of what is obvious and what is not. The rest of us need a little more hand-holding.
185 – change “and observation periods” to “with observation periods.”
199 – Remove “Remarkably, the”, just start with “Mangshan pit vipers allocated…”. These qualifying adjectives are really not necessary or appropriate.
200 – change “in a balance manner” to “in approximately equal proportions.”
217-219 – This section essentially says, “of tested factors, only temperature and illumination had a statistically significant effect on arboreality (stats); humidity, hour, and season were not significant.” Since that sums up your results nicely, I’m not sure that Table 1 is necessary at all, or Tables 2-4 for the same reason. You might offer them as supplemental material, but even then it’s an awful lot of real estate to devote to a relatively simple analysis that is already summarized efficiently in the text.
220-224 – Again you’re falling here into the trap of presenting statistics independently of their meaning. You’re going to lose many readers by talking about your analysis finding a lower residual sum of squares. Instead, what did it find in terms of snake activity – the thing that you are actually testing?
246 – what is Leopoldamys edwardsi? I shouldn’t have to look up elsewhere that it’s a rodent. Is this a common species at the study site? Are Mangshan pit vipers typically predators of small mammals? If snakes adjust their activity to mirror that of their prey, that’s a REALLY interesting result. You need to paint a fuller picture here to set up that idea, because it’s an important one.
284 – remove the word “And.”
287 – remove “Fascinatingly.”
287-295 – this makes total sense, as I expect temperature drops in autumn so snakes spend more time basking to warm up. You should include seasonal temperature data to clarify and emphasize this point.
312-322 – Ah, here is the temperature data I was looking for. I would combine this with the previous paragraph, or at least present this one first. They make essentially the same point.
355 – what was the “expected” habitat? Why? You haven’t said anything about P. mangshanensis’s body shape, or that of highly arboreal snakes.
378 – change “vari-ables” to “variables.”
379 – change “maxi-mum” to “maximum.”
383 – remove “…in our analysis” (redundant).
391 – I don’t follow this argument about canopy providing light. If anything, wouldn’t increased canopy cover allow LESS light, with gaps (like treefalls) having higher light? Did snakes seem to use such gaps?
Alternatively, are you talking about snakes being higher up in the canopy for access to sunlight, as opposed to close to the ground? If that’s the case that’s a very interesting possibility, but it should be clarified and explored quite a bit. Did you (or can you) quantify light penetration with height? In other words… how much light penetration is there on the ground, vs. at different heights?
For that matter, for all the emphasis on arboreality, I don’t see any data on actual height, which in retrospect seems like a glaring omission. How high did snakes actually climb to bask? 1m? 10? 30? What counted as arboreal… what if they were on a branch just above the ground?
401 – change “thses” to “these.”
402-403 – what geographic, climatic, and other resources? Why would these vary with season?
Figure 1 – There is nothing in the legend to indicate what A-D means. What do these letters refer to?
The thin red line outlining the Reserve is very hard to see in a printed version. I’d recommend using a heavier black line.
I also note that in the figure captions, you refer to them as “pitvipers”, one word. The rest of the manuscript uses “pit vipers”, the preferred form.
Figure 2 – I would simplify this caption quite a bit, to “Behaviors of Mangshan pitvipers in their natural habitat, photographs by Bing Zhang. A) arboreal resting, B) arboreal locomotion, C) terrestrial resting, D) terrestrial locomotion.
Figure 3 – This graph contains extremely useful data, but is very difficult to read. Axis labels are so small as to be illegible. I would simplify the x-axis time labels like in Figure 4.
I understand that you are illustrating how consistent activity is, with so little individual variation. And that’s a terrific point, but something like a column graph (of mean activity time across all snakes, with whiskers for Confidence Intervals) would probably make it more efficiently.
Figure 4 – This figure makes a great point (snake activity seems to correlate with rat activity), maybe the most interesting of your findings. Unfortunately, it’s again a bit hard to read visually. Grouping some of these data into columns, or maybe an alternative format like a line graph, might be a more visually-effective way to illustrate the point. It’s hard to make a specific suggestion as I don’t follow what “movement frequency” is in this context or exactly what data this graph is based on.
Figure 5 – I again can’t read these axis labels AT ALL.
Figure 6 – A nice visual summary of snake activity by season is very useful. However, I’m not sure this is the most effective graph type to do it. A column or pie chart seem like they would make the same case more effectively and intuitively. Graphs are a visual medium, so ease of interpretation is key… all else being equal, simpler is usually better.
Figure 7 – Now THIS is a nice graph. Clean, simple, easy to read. I immediately understand what each axis is, exactly what data are being shown, and what the overall “story” is.
I would probably lower your y-axis range a bit, doesn’t look like there’s anything over 30. The legend should explicitly state that points are (I assume) means from all individuals, and whether those whiskers are your minimum/maximum values, or 95% CIs.
Figure 8 – another great graph. Again, you just need to explicitly state what your data points are in the legend.
Comments on the Quality of English Language
Author Response
Dear Reviewer,
Thank you for your valuable feedback on our paper. We appreciate your comments and suggestions for improving our manuscript.
We agree that the title could be improved to better reflect the remote-sensing approach used in our study. Your suggestion of "Activity patterns of the critically endangered Mangshan pit viper determined using remote monitoring" or a similar title is excellent, and we will definitely consider incorporating it into our revised manuscript.
Regarding your concern about our focus on statistical results, we completely agree that the biological patterns we are testing should be the main focus of the paper. In our revision, we have made revisions based on the feedback provided by the reviewers, sections to clearly emphasize the biological significance of each analysis and what the results mean in the context of Mangshan pit vipers. Statistics will be presented as a tool to support our findings, rather than the primary focus.
We also appreciate your suggestions for improving clarity and readability. We will carefully consider each of your comments and make the necessary adjustments to enhance the overall quality of the manuscript.
Line 24 – Change “Focuses on…” to “This study focuses on…”
Thanks for your careful checks. Based on your feedback, we have made the necessary corrections in line 26.
35 – I would probably rephrase this a bit. It’s not that snakes from terrestrial and arboreal habitats used different activity patterns, it seems like snakes used terrestrial and arboreal habitats under different environmental conditions.
Thank you very much for your thoughtful feedback. We deeply appreciate your guidance and have carefully revised the manuscript based on your valuable suggestion. The adjustment is now reflected in lines 38-39 of the revision.
36 and throughout – Really, I usually see the term “habitat usage” used when quantifying the actual macrohabitat or microhabitat… substrate, structural features like rocks and logs, canopy coverage, etc. I think the authors are clear enough about what they’re doing here that the term is fine, but I would probably just change it to “arboreality” to avoid any ambiguity.
This also makes me wonder if the authors don’t have enough data for a subsequent study of microhabitat usage by P. mangshanensis. There’s probably an awful lot of microhabitat data in their snake videos.
Thank you for your careful checks. We apologize for the omit on our part. Based on your comments, we have made corrections to ensure consistency in terminology throughout the manuscript.
Regarding your suggestion on microhabitat usage, we appreciate your insight. However, a study on the microhabitat of P. mangshanensis was previously conducted by our co-author, Zhang B, in 2020 using traditional methods. The results from that study are published in PeerJ (Zhang B, Wu B, Yang D, Tao X, Zhang M, Hu S, Chen J, Zheng M. Habitat association in the critically endangered Mangshan pit viper (Protobothrops mangshanensis), a species endemic to China. PeerJ. 2020 Jul 1;8:e9439). In our current study, the data obtained from the monitors did not reveal any significant differences from the previous findings. Therefore, we have not included a detailed discussion on the impact of microhabitats in this paper. We hope this clarifies our approach and rationale behind the current study.
50 and throughout – The word “diurnal” means “during the daytime” (as opposed to nocturnal, at night). I assume here the authors are talking about an animal’s 24-hour cycle, so “diel” is the appropriate technical term. Something less formal like “temporal” or “daily” would also be fine.
Thank you for pointing out this terminology issue. We appreciate your feedback and have revised the manuscript to consistently use the term "diel" to refer to the animal's 24-hour cycle in the revision
54 – Italicize Protobothrops mangshanensis.
We feel sorry for our carelessness. In our resubmitted manuscript, the typo is revised in line 58. Thanks for your correction.
71-75 – You should briefly explain what that Five-Year Plan is and what its goals are. Most international readers will be unfamiliar with it and will not be able to read the linked website.
Thank you for your feedback. We appreciate your suggestion to clarify the Five-Year Plan for our international readers. In lines 79-93 of the revised manuscript, we have provided a description of this initiative by the Chinese government, which aims to restore the ecology and population of rare and endangered animals. We believe this addition will enhance the readability and understanding of our paper for a wider audience.
89 – Again diel and not diurnal; you’re also (as above) not really quantifying habitat utilization. I would specify it as just “arboreality” or something similar.
Thank you for your feedback. We have made the necessary modifications in line 108.
96-106 – Your study site may be well described in Zhang 2020, but you should explain a little more about it here. There are several forest types, but is it generally dry forest, temperate forest, rainforest? At least there should be temperature and precipitation range. You want your readers to have a general sense of the area, as very very few will be previously familiar with the Reserve.
Thank you for your suggestion. We agree that providing more information about the study site is important for readers to understand the context of our research. In lines 119-123 of the revised manuscript, we have added a brief description of the natural environment of Mangshan Nature Reserve, including its forest types, temperature, and precipitation range. This additional information will give readers a general sense of the area and aid in their understanding of our study. We appreciate your guidance in improving the clarity and comprehensibility of our paper.
116 – I am not sure what “instantaneous scan sampling” and “all occurrence observation methods” are. You should link these terms to your actual methods, though I’m not sure that they’re entirely helpful or necessary at all.
Thank you for your feedback. We understand your confusion regarding the terms "instantaneous scan sampling" and "all occurrence observation methods". While we utilized infrared monitoring instruments in our study, the fundamental approach we employed to measure animal movement distances and time intervals is indeed based on these two methods, as they are known in the literature. To clarify our methodology and provide context for these terms, we have included a description of "instantaneous scan sampling" and "all occurrence observation methods" in lines 142-145 of the revised manuscript. Thank you for helping us improve the clarity of our manuscript.
126 – Using infrared cameras is interesting. Do snakes show up well using thermal imaging, being ectotherms? Or were they used just to allow monitoring in low-light conditions at night?
Thank you for your question regarding the use of infrared cameras to monitor snakes. It is indeed an interesting point. In response to your inquiry, it should be noted that snakes, as cold-blooded animals, do not generate their own body heat and therefore may not show up as distinctly on thermal imaging as warm-blooded animals. However, in our study, we primarily used infrared surveillance cameras to facilitate monitoring in low-light conditions at night.
It's important to mention that infrared surveillance becomes ineffective for most observations during the period from one hour after sunset to one hour before sunrise, roughly around 8:30 PM to 5:00 AM local time each day. During this time, the snakes may become difficult to detect due to the limitations of the technology.
To address this challenge, we have adopted a specific definition for stationary states in our study. If the observed target's position and posture remain unchanged from the moment it becomes completely invisible due to low light conditions until the next time it becomes visible again, we consider it as stationary and without movement.
We have included this information in lines 171-178 of the revised manuscript to provide clarity on the use and limitations of infrared cameras in our study. Thank you for raising this important point, and we hope this explanation addresses your question.
135 – From the way it’s used later, “displacement” needs some clarification here. When you are measuring displacement… are you quantifying ALL movements over the 24 hour period, or straight-line displacement from the original location? For example… if a snake moves 5m to the east, and stops; then later it moves 2m to the west (back towards its original starting point), and stops… that is a total of 7m movement, correct? Not a linear displacement of 3m from the original starting point. This is fine, but should be clarified. A very simple explanation is fine, but “document their displacement” is too vague.
Thank you for pointing out the need for clarification regarding "displacement" in our study. Your question raises an important distinction that we should have made more explicit in our original manuscript.
In our study, when measuring the displacement of snakes, we quantified all movements over the 24-hour period, following the actual movement trajectory of the snakes as closely as possible. This means that we measured the total distance traveled, rather than the straight-line displacement from the original location, to clarify this in the manuscript, we have added a description in lines 164-166.
To illustrate with your example, if a snake moves 5m to the east and stops, and then later moves 2m to the west (back towards its original starting point) and stops, we would consider that a total of 7m of movement.
Thank you for your valuable feedback, which has helped us improve the clarity and accuracy of our methods description.
147 – I cannot understand why August 7th data would be excluded. Why not just assign that day to Autum?
Thank you for raising this question about the exclusion of August 7th data. The reason for excluding this day's data is related to the specific timing of the "Beginning of Autumn" in Chinese meteorology.
The China Meteorological Administration designates the "Beginning of Autumn" based on a precise time point, which in 2021 was at 14:53:48 on August 7th. This time point marks the transition from summer to autumn, rather than considering the entire day of August 7th as part of autumn. We interpret this time as a fixed point in the Earth's orbit around the sun, and when the Earth passes this point, it enters autumn.
Given that on August 7th, there were 15 hours in summer and 9 hours in autumn, the data for that day does not represent a full 24-hour period in either season. Therefore, to maintain the integrity and accuracy of our seasonal data analysis, we decided to exclude the data from August 7th.
We have clarified this point in lines 187-189 of the revised manuscript, explaining the rationale for excluding the August 7th data.
151 – Frequency of movement is different from displacement (distance of movement), and not mentioned in the Methods. Are you talking about the number of movements, or whether the snake moved at all? You really need to be very careful, specific, and consistent with the terminology of your methodology.
Thank you for pointing out the need for clarity regarding the terminology used in our methodology, specifically related to the frequency of movement. Your feedback is valuable in helping us ensure consistency and precision in our descriptions.
In response to your question, the frequency of movement in our study refers to the ratio of the duration of movement to the total recording time. We have calculated this by dividing the time spent moving by the snake during the observation period by the total recording time.
The reason we chose to use frequency instead of direct time measurements is to facilitate comparison with rodent movement data provided by other researchers. These rodent movement frequencies were calculated based on the number of movement photos captured by triggered infrared cameras within a specific time frame (1 hour), divided by the total number of photos taken of the species during that period. Although the camera's shooting speed is fixed, and thus, to some extent, reflects time, the rodent data was not directly based on time measurements. To align our snake movement data with this approach and enable meaningful comparisons, we converted snake movement into frequencies as well.
We appreciate your feedback and have revised the manuscript to clarify this aspect of our methodology in lines 194-198. This clarification will help readers better understand our approach and the rationale behind it. Thank you for your careful review and suggestions for improvement.
159-164 – This is a good example of what I mentioned earlier about emphasizing statistics vs. their biological meaning. These models are well-designed, and stating their parameters is probably useful to those interested in similar methods. But what do they MEAN biologically? What does a main-effects vs. an interaction-effects model imply is important to a snake’s activity? I urge the authors to carefully consider that many readers will likely be more familiar with and more interested in animal-natural-history aspects, so think about presenting your methods and results with that perspective in mind. “In order to determine x, we used analysis y.”
Thank you for emphasizing the importance of connecting statistical methods to their biological meaning. We fully agree that it is crucial to present our methods and results in a way that is accessible and meaningful to a wide range of readers, including those who are more familiar with animal natural history.
In response to your feedback, we have revised the manuscript to better explain the biological implications of our statistical models. Specifically, we have clarified that the main-effects model explores the individual effects of environmental factors on snake activity, while the interaction-effects model investigates how these factors combine to influence snake behavior.
We have made modifications in lines 199-201,232~238 to explicitly state that our goal was to examine the impact of environmental factors on the species' habitat use patterns. By employing these models, we aimed to gain insights into how various environmental conditions shape the snakes' activity and habitat preferences.
168-180 – Same comment as above. Ex. On line 168… to do WHAT with movement and habitat data? What exactly will these analyses show? Ex. line 174… for example, I might say something like “To determine the effects of environmental conditions on arboreality by snakes, we used Multinomial Logistic Regression within a GLM framework… etc.” This is much better than saying “we used this approach for these data” Instead it helps orient readers to understand exactly what is being tested, what that analysis does, and what they can expect the results to mean.
I again urge the authors to consider this for all their analyses. Remember, NOBODY understands this stuff as well as you do. When you’re so close to the study for so long, it’s easy to lose sight of what is obvious and what is not. The rest of us need a little more hand-holding.
Thank you for your valuable feedback on improving the clarity of our methods and objectives. We appreciate your emphasis on helping readers understand the purpose and implications of our analyses.
Thank you for emphasizing the importance of connecting statistical methods to their biological meaning again, we have made specific revisions to the manuscript. On lines 223-224, we have clarified that the purpose of our analysis is to determine the effects of environmental conditions on the snakes' behavior. Furthermore, on lines 232-238, we have elaborated that a significant p-value in our models indicates that the corresponding environmental factor (temperature, humidity, or illumination) has an impact on the species' behavior (habitat use patterns, movement, or activity patterns).
We agree that it is essential to provide a clear and accessible narrative for readers, guiding them through the analyses and explaining the significance of the results. Your feedback has helped us to refine our presentation, ensuring that the biological implications of our statistical analyses are more evident.
185 – change “and observation periods” to “with observation periods.”
Thanks for your careful checks. We are sorry for our carelessness. Meanwhile, according to the requirements of other reviewers, we need to rearrange the sentence positions. And based on your comments, we have made the corrections in 244~245.
199 – Remove “Remarkably, the”, just start with “Mangshan pit vipers allocated…”. These qualifying adjectives are really not necessary or appropriate.
Thank you for the reviewer's suggestion. We have removed "Remarkably, the" from the sentence as advised. This change has been implemented in line 259 of the revision
200 – change “in a balance manner” to “in approximately equal proportions.”
Thank you for the reviewer's recommendation. We have modified the phrase "in a balanced manner" to "in approximately equal proportions" as suggested. This alteration has been made in line 260 of the revision for clarity and precision.
217-219 – This section essentially says, “of tested factors, only temperature and illumination had a statistically significant effect on arboreality (stats); humidity, hour, and season were not significant.” Since that sums up your results nicely, I’m not sure that Table 1 is necessary at all, or Tables 2-4 for the same reason. You might offer them as supplemental material, but even then it’s an awful lot of real estate to devote to a relatively simple analysis that is already summarized efficiently in the text.
Thank you for the reviewer's feedback on the use of tables in our manuscript. While we appreciate the suggestion to offer Tables 1, 2, 3, and 4 as supplemental material, we believe that these tables are important to include in the main body of the text for several reasons.
Firstly, the tables provide detailed numerical data that supports our conclusions regarding the statistically significant effects of temperature and illumination on arboreality. These numbers give readers a more precise understanding of the magnitude and direction of these effects, which is not fully captured by the textual summary.
Secondly, while the text provides a general overview of the results, the tables offer a more comprehensive and specific presentation of the data. This allows interested readers to explore the results more deeply and draw their own conclusions based on the detailed information provided.
Lastly, we believe that including these tables in the main text enhances the transparency and reproducibility of our research. By providing the raw data and statistical outputs, we enable other researchers to verify our findings and build upon them in their own studies.
Therefore, we respectfully request to keep Tables 1, 2, 3, and 4 in the main body of the text. We believe that they add value to our manuscript and contribute to a more complete understanding of our research findings.
220-224 – Again you’re falling here into the trap of presenting statistics independently of their meaning. You’re going to lose many readers by talking about your analysis finding a lower residual sum of squares. Instead, what did it find in terms of snake activity – the thing that you are actually testing?
Thank you, reviewer, for pointing out this important issue. We agree that presenting statistics without connecting them to the actual research question and the meaning behind the data can be confusing and may lose readers. In response to your feedback, we have deleted the description in lines 280-284, and the explanation needed here has already been provided in lines 232-238.
246 – what is Leopoldamys edwardsi? I shouldn’t have to look up elsewhere that it’s a rodent. Is this a common species at the study site? Are Mangshan pit vipers typically predators of small mammals? If snakes adjust their activity to mirror that of their prey, that’s a REALLY interesting result. You need to paint a fuller picture here to set up that idea, because it’s an important one.
Thank you for the reviewer's comments. We have incorporated additional information in lines 309-311 to clarify that Leopoldamys edwardsi is the primary rodent species at the study site and one of the main prey items for Mangshan pit vipers. Furthermore, in the discussion section from lines 409-504, we have expanded on the idea that snakes adjust their activity to mirror that of their prey, highlighting the significance of this behavior and discussing its ecological implications. We appreciate the reviewer's feedback, which has helped us to improve the clarity and completeness of our manuscript.
284 – remove the word “And.”
We feel sorry for our carelessness. In our resubmitted manuscript, the typo is revised in line 351. Thanks for your correction.
287 – remove “Fascinatingly.”
We apologize for our negligence. In our revised manuscript, the typo has been corrected in line 365. We are grateful for your assistance in identifying and rectifying this error.
287-295 – this makes total sense, as I expect temperature drops in autumn so snakes spend more time basking to warm up. You should include seasonal temperature data to clarify and emphasize this point.
Thank you for the reviewer's feedback. Taking into account your next suggestion as well, we have rearranged the paragraphs and moved the discussion about temperature to a more prominent position. And in lines 418-419, we have included information about the average daily temperatures recorded during our study period in summer and autumn.
312-322 – Ah, here is the temperature data I was looking for. I would combine this with the previous paragraph, or at least present this one first. They make essentially the same point.
Thank you for the reviewer's feedback. Considering your subsequent suggestion, we have reorganized the text and relocated the discussion on temperature to a more visible spot, specifically to lines 354-364.
355 – what was the “expected” habitat? Why? You haven’t said anything about P. mangshanensis’s body shape, or that of highly arboreal snakes.
Thank you for the reviewer's feedback. We have added a general description of P. mangshanensis and its presumed terrestrial behavior in lines 443-447. This information helps to clarify the "expected" habitat of this species and provides context for why it was initially thought to be primarily terrestrial.
378 – change “vari-ables” to “variables.”
We're sorry for the typo in our original submission. In the revised version, "vari-ables" has been corrected to "variables" in line 473. Thank you for pointing it out.
379 – change “maxi-mum” to “maximum.”
We regret the typographical error in our initial manuscript. The word "maxi-mum" has now been fixed to "maximum" in line 474 of the revised manuscript. Your attention to detail is appreciated.
383 – remove “…in our analysis” (redundant).
We acknowledge the redundancy of "...in our analysis" and have removed it as suggested in line 478 of the updated manuscript. Your feedback helps us improve the clarity of our writing. Thank you.
391 – I don’t follow this argument about canopy providing light. If anything, wouldn’t increased canopy cover allow LESS light, with gaps (like treefalls) having higher light? Did snakes seem to use such gaps?
Alternatively, are you talking about snakes being higher up in the canopy for access to sunlight, as opposed to close to the ground? If that’s the case that’s a very interesting possibility, but it should be clarified and explored quite a bit. Did you (or can you) quantify light penetration with height? In other words… how much light penetration is there on the ground, vs. at different heights?
For that matter, for all the emphasis on arboreality, I don’t see any data on actual height, which in retrospect seems like a glaring omission. How high did snakes actually climb to bask? 1m? 10? 30? What counted as arboreal… what if they were on a branch just above the ground?
Thank you for your valuable feedback, reviewer. You are correct in pointing out that my previous explanation regarding canopy providing light might have been unclear. What I meant to convey was that snakes may seek higher positions in the canopy to access sunlight, as opposed to staying close to the ground where light penetration might be limited.
Although we did not directly quantify the relationship between light penetration and height in our study, we have added references (42, 43) in lines 485-487 that explore light intensity and its vertical distribution in forests. These studies provide insights into how light varies with height in a forest canopy.
Regarding your question about the actual height snakes climbed to bask, we have included additional information in the methods section (lines 179-181) clarifying how we defined arboreality in this study. While we currently do not have direct data on the specific heights snakes climbed, our observations and inferences are based on their behavior and the habitats they were found in.
In the discussion section, we have revised the text in lines 491~494 to reflect that our conclusions are based on inferences from the study results, acknowledging the lack of direct data to support these findings. We appreciate your feedback, which has helped us clarify our arguments and strengthen the manuscript.
401 – change “thses” to “these.”
We apologize for the typo and thank you for pointing it out. We have made the correction from "thses" to "these" in line 501 of the revised manuscript.
402-403 – what geographic, climatic, and other resources? Why would these vary with season?
Thank you for your inquiry. In the revision, I have added more details in lines 510-514 to elaborate on how species adapt their behaviors in response to environmental factors. Specifically, they may seek suitable temperatures, search for water sources, and avoid excessive humidity, among other behaviors. These adaptive behaviors reflect the species' adaptability and tolerance to their environment. Geographic and climatic conditions, as well as the availability of other resources, can vary with the seasons, influencing the species' behavioral patterns. For instance, changes in temperature and precipitation can affect where animals forage or nest, and seasonal variations in food availability may drive migratory patterns.
Figure 1 – There is nothing in the legend to indicate what A-D means. What do these letters refer to?
The thin red line outlining the Reserve is very hard to see in a printed version. I’d recommend using a heavier black line.
I also note that in the figure captions, you refer to them as “pitvipers”, one word. The rest of the manuscript uses “pit vipers”, the preferred form.
Thank you for your suggestion. We appreciate your attention to detail and helpful suggestions. We have made the necessary corrections to ensure consistency in the manuscript. Specifically, we have unified the term "pit vipers" throughout the text to align with the preferred form.
Regarding the legend for A-D, we have now included annotations on the map to clarify what these letters represent. They correspond to the photos of the habitats surrounding the species we discovered.
As for the visibility of the Reserve boundary, we agree that the thin red line was not sufficiently visible in the printed version. However, upon testing, we found that a heavier black line was not ideal against the dark green background of the map. Instead, we have opted to change the boundary line to white, which may provide better contrast and visibility for readers.
Figure 2 – I would simplify this caption quite a bit, to “Behaviors of Mangshan pitvipers in their natural habitat, photographs by Bing Zhang. A) arboreal resting, B) arboreal locomotion, C) terrestrial resting, D) terrestrial locomotion.
Thank you for your decision and constructive comments on my manuscript.Thank you for helping me improve the clarity and readability of my manuscript.
Figure 3 – This graph contains extremely useful data, but is very difficult to read. Axis labels are so small as to be illegible. I would simplify the x-axis time labels like in Figure 4.
I understand that you are illustrating how consistent activity is, with so little individual variation. And that’s a terrific point, but something like a column graph (of mean activity time across all snakes, with whiskers for Confidence Intervals) would probably make it more efficiently.
Thank you for your valuable feedback on our manuscript. We appreciate your suggestions for improving the readability of our graphs and the clarity of our data presentation.
Regarding the graph you mentioned, we agree that the axis labels were too small and difficult to read. We have now increased the font size to ensure legibility.
We understand your suggestion to use a column graph to illustrate the mean activity time across all snakes with confidence intervals. However, upon experimentation, we found that such a representation would be identical to our current Figure 6. Our primary objective is to showcase the consistency of activity among individual snakes, and we believe that the current representation effectively conveys this point.
To further enhance the visualization and provide a more comprehensive understanding of the data, we have decided to replace the existing graph with a heatmap. This heatmap represents the activity level of snakes at each stage using a color gradient, allowing for a quick and intuitive visualization of activity patterns. While this representation does not directly display standard deviations, we have included the complete dataset, including standard deviations, in the supplementary data file for interested readers to explore further.
Figure 4 – This figure makes a great point (snake activity seems to correlate with rat activity), maybe the most interesting of your findings. Unfortunately, it’s again a bit hard to read visually. Grouping some of these data into columns, or maybe an alternative format like a line graph, might be a more visually-effective way to illustrate the point. It’s hard to make a specific suggestion as I don’t follow what “movement frequency” is in this context or exactly what data this graph is based on.
Thank you for your feedback on our figure. We appreciate your comments on the visual readability and your suggestion to explore alternative formats. We have taken your advice to heart and made some significant improvements to the figure.
Firstly, for further clarity on the calculation of movement frequency, we have provided a detailed explanation in the text (lines 194-198), as you mentioned. This explanation outlines the methodology used to determine the frequency of movements and serves as a reference for readers to fully comprehend the data presented in the figure.
In response to your suggestion, we have revised the figure to incorporate both a bar graph and a line graph. The bar graph clearly illustrates the movement frequency of snakes at different time points, while the line graph traces the rats at different times. This dual representation allows for a more comprehensive understanding of the data.
Additionally, we have simplified the x-axis to enhance readability. The revised figure now presents the data in a more visually appealing and easily understandable format.
Thank you once again for your valuable feedback, which has helped us refine our manuscript and improve the communication of our research.
Figure 5 – I again can’t read these axis labels AT ALL.
Thank you for pointing out the issue with the axis labels. We apologize for the inconvenience caused. We have now made significant improvements to the figure (which is now Figure 6) based on your feedback.
Specifically, we have simplified the x-axis labels to ensure better readability. The new labels are larger, clearer, and more straightforward, making it easier for readers to interpret the data presented in the graph.
We believe these changes address the concerns you raised and significantly enhance the readability and interpretability of our figure. Thank you again for your valuable feedback, which has helped us refine our manuscript and improve the communication of our research.
Figure 6 – A nice visual summary of snake activity by season is very useful. However, I’m not sure this is the most effective graph type to do it. A column or pie chart seem like they would make the same case more effectively and intuitively. Graphs are a visual medium, so ease of interpretation is key… all else being equal, simpler is usually better.
Thank you for your feedback on our visual representation of snake activity by season. We appreciate your suggestion to explore alternative graph types for clearer and more intuitive data presentation.
In response to your comments, we have revised the figure to a contingency stacked bar chart. This chart type allows for a more straightforward comparison of snake activity across different seasons. However adding error bars to this type of chart can be technically challenging in commonly used graphing software such as Excel, Prism, and R ggplot.
Due to these limitations, we have decided to include the standard deviations in the supplementary data instead of directly on the graph. This approach ensures that readers can still access the necessary information to assess the variability within each season while maintaining a clean and uncluttered visual representation.
We believe this revision improves the readability and interpretability of our figure, making it easier for readers to understand the seasonal patterns of snake activity. Thank you again for your valuable input, which has helped us refine our data visualization and enhance the communication of our research findings.
Figure 7 – Now THIS is a nice graph. Clean, simple, easy to read. I immediately understand what each axis is, exactly what data are being shown, and what the overall “story” is.
I would probably lower your y-axis range a bit, doesn’t look like there’s anything over 30. The legend should explicitly state that points are (I assume) means from all individuals, and whether those whiskers are your minimum/maximum values, or 95% CIs.
Thank you for your positive feedback on our graph. In response to your comments, we have made some further adjustments to improve the clarity and accuracy of the graph. Firstly, we have lowered the y-axis range as you suggested, adjusting the upper limit to 30 since there are no data points above this value. This adjustment helps to focus the viewer's attention on the relevant data range and improves the readability of the graph.
Additionally, we have clarified that the whiskers represent standard deviations. This information is now included in the legend of all graphs with whiskers to ensure clarity and consistency.
We appreciate your valuable input, which has helped us further refine our data visualization. Thank you for your continued support and feedback, which are crucial in helping us communicate our research findings effectively
Figure 8 – another great graph. Again, you just need to explicitly state what your data points are in the legend.
Thank you for your feedback on our graph. We are pleased that you find it satisfactory, and we appreciate your suggestion to clarify the data points in the legend.
To address your comment, we have added a description to the graph's title, indicating that the bars represent mean values and the whiskers denote standard deviations.
Thank you again for your thoughtful review. Your feedback has been invaluable in helping us refine our work. We look forward to submitting a revised version that incorporates your suggestions and better communicates our findings.
Best wish

Reviewer 2 Report
Comments and Suggestions for Authors
This study reports original data on the abiotic and biotic factors that influence the daily activity rhythm of a critically endangered snake species, endemic to China, using non-invasive methods. It deserves interest in publication.
However, there are some points that leave me perplexed and make the manuscript not yet ready for publication:
1) In the main text, the bibliographic citations start from papers [7-10], not from paper [1]. Furthermore, they do not always seem to be consistent with what is stated in the text. For example, at lines 57-58, the authors state: “…Listed as "globally endangered" on the IUCN Red List and designated a first-class protected animal in China[2]“ but the source [2] describes the crystal structure of an Arg49 phospholipase A2 homolog isolated from Z. mangshanensis venom; it does not report the conservation status of the species. Similarly, at line 59, the citation [4] is a paper on long term individual identification using head patch pattern and does not deal with population density; on the other hand, data concerning populations with fewer than 500 individuals can be found in Zhang et al, 2020 (bibliographic citation [5]). Therefore, I would suggest carrying out a careful check of the cited references.
2) Different parts of the text (notably the results, including figure captions) should be reorganized trying to use more concise language and removing sentences that are more appropriate to other sections of the ms (e.g. see lines 186-188; line 217; 246 -249).
3) A weakness of the study, underlined also by the authors, is how much the fact of not having had the possibility of distinguishing the two sexes could have an influence in terms of management of the species. I believe this aspect needs to be better explored
My detailed comments are in the attached pdf

Author Response
Dear Reviewer,
Thank you for your detailed feedback on our manuscript. We appreciate your constructive criticism and have addressed each of your points as follows:
1) We apologize for the confusion caused by the bibliographic citations. During the revision of the introduction, we accidentally removed the cross-references, leading to a mismatch in citation numbers. We have now corrected this error and renumbered the citations accordingly. Additionally, we have carefully checked all citations to ensure they are consistent with the text and accurately reflect the content of the referenced papers.
2) Thank you for your suggestions regarding the reorganization and language usage in the text. We have carefully revised the manuscript, particularly the results section and figure captions, to ensure concise and clear language. Sentences that were more appropriate for other sections have been moved or removed, as per your recommendations. This will be addressed individually below.
3) We appreciate your pointing out the weakness of our study regarding the lack of distinction between the two sexes. In response, we have added a discussion in lines 520-526, where discussed the importance of sexual dimorphism in snake research.
Please write in italics
We feel sorry for our carelessness. In our resubmitted manuscript, the typo is revised in line 58. Thanks for your correction.
In my opinion, since the snakes are not motionless, it is more appropriate to say "...cameras near the paths usually followed by the snakes so that they can film them"
Thank you for your insightful and valuable suggestion. In response to your feedback, we have revised the text in lines 148-149 of the manuscript to reflect a more accurate and clear description of the camera placement.
This sentence is unclear
Thank you for pointing out the unclear sentence. We appreciate your feedback and have taken steps to address the issue. In response to your comment, we have provided a detailed explanation in lines 187-189 of the manuscript to clarify the unclear sentence.
start
We feel sorry for our carelessness. In our resubmitted manuscript, the typo is revised in line 146.
Please better specify how the frequency of movement was measured
Thank you for your feedback regarding the specification of how the frequency of movement was measured. We appreciate your comment and have made modifications in lines 194-198 of the manuscript to clarify this aspect. We have included detailed explanations on how both the movement frequency of snakes and rodents was calculated.
In addition, using
We feel sorry for our carelessness. In our resubmitted manuscript in line 223, the typo is revised. Thanks for your correction.
is part should be reorganized. Move this sentence after "However, ...undetermined", changing it: "observation period was from 3 to 6 days for snake
Thank you for your suggestion to reorganize this part of the manuscript. We appreciate your feedback and have accordingly made modifications in lines 243-249, as you recommended.
arboreal (Fig. 2A-B) and terrestrial (Fig. 2C-D) habitats
Thank you for your feedback. We appreciate your comments and have carefully considered them in our revisions. In response to your suggestion, we have made modifications in lines 247-248 of the manuscript.
Diverse, I believe this noun is unnecessary and perhaps a little bit anthropomorphized and I believe this noun also is unnecessary previous title of Figure 2.
We appreciate your comments and have taken them into consideration for our revisions. In response to your suggestions, we have rephrased the title of Figure 2 to "Behaviors of Mangshan pitvipers in their natural habitat, photographs by Bing Zhang." In lines 252~253. This new title provides a clearer and more direct description of the figure's content.
“Habitats”, “am”,“In”
We apologize for our negligence. In our revised manuscript, the typo has been corrected in line 260, line 261, line 269 and line 270. And the extra expression has been removed in line 278.
GLM
We appreciate your comments and have taken it in line 279.
I would suggest writing the caption more clearly, starting with "Generalized Linear model (GLM) based on ..."
Thank you for your valuable suggestion to clarify the caption of Table 1. We appreciate your feedback and have taken it into consideration for our revisions. In the resubmitted manuscript, we have rewritten the caption of Table 1 in lines 276-279, following your recommendation.
Please insert space between "CI:" and "Confidence interval"
We apologize for our negligence. In our revised manuscript, the typo has been corrected in line 292.
Although intuitive, I think it is better to specify in the caption: "each individual is represented by a different color"
Thank you for your valuable feedback on our manuscript. We appreciate your suggestions for improving the readability of our graphs and the clarity of our data presentation.
To further enhance the visualization and provide a more comprehensive understanding of the data, we have decided to replace the existing graph with a heatmap. This heatmap represents the activity level of snakes at each stage using a color gradient, allowing for a quick and intuitive visualization of activity patterns. While this representation does not directly display standard deviations, we have included the complete dataset, including standard deviations, in the supplementary data file for interested readers to explore further.
Move this sentence in Discussion. In addition, please tie the source where this data was published or, in the negative, specify that they were upublished (data (personal observation) .
Thank you for your feedback. We appreciate your comments and have incorporated the necessary changes in our revised manuscript.
As per your suggestion, we have moved the sentence to the Discussion section, specifically to lines 503-507 in the resubmitted manuscript. This relocation ensures that the information is presented in a more logical and coherent manner within the overall narrative of the paper.
Additionally, I have identified the published source for the data mentioned and have added it as the 24th reference in the bibliography. This provides readers with a direct link to the original source, enhancing the transparency and credibility of our work.
Thank you once again for your valuable input.
Please see my comments about Table 1. In addition, please don't use small caps
We apologize for our negligence. In our revised manuscript, the typo has been corrected, and we have rewritten the caption of Table 2 in lines 331~334.
Please insert space between "CI:" and "Confidence interval"
We apologize for our negligence. In our revised manuscript, the typo has been corrected in line 337.
Please see my comments about Table 1
Thank you for your valuable suggestion to clarify the caption of Table 3. We appreciate your feedback and have taken it into consideration for our revisions. In the resubmitted manuscript, we have rewritten the caption of Table 3 in lines 339~341, following your recommendation.
Please insert space between "CI:" and "Confidence interval"
We apologize for our negligence. In our revised manuscript, the typo has been corrected in line 344.
I find that this sentence, as well as the sentence at Lines 282-283. also should be moved into Discussion
Thank you for your feedback. We appreciate your comments and have taken them into consideration for our revisions. As your suggestion, we have moved the sentence from the original location to the discussion section, specifically to lines 439-441 in the resubmitted manuscript.
There is something that is not clear to me: this caption is identical to that of table 1
Thank you for pointing out this issue. We appreciate your feedback and have taken it into consideration for our revisions. In the resubmitted manuscript, we have modified the caption of Table 4 at lines 380-381.
Please specify (it' s also fine in the caption) what the y-axis and x-axis report
Thank you for pointing out our oversight. We apologize for not including clear labels for the axes in our figure. We appreciate your feedback and have rectified this issue in the resubmitted manuscript. In the revision, we have updated Figure 6 to include specific labels for both the x-axis and y-axis.
About the temperature effects on activity levels, we observed ...
Thank you for your feedback on our manuscript.we have carefully revised the relevant section in response to your comments. In the revised manuscript, at line 335.
was
We apologize for our negligence. In our revised manuscript, the typo has been corrected in line 357.
“Relationship” and delete
Thank you for your feedback on our manuscript. As per your suggestion, we have modified the title of Figure 5 in the manuscript. This change has been implemented in lines 368-369 of the revised manuscript.
Please don't use small caps
I appreciate your feedback on our manuscript. Taking your suggestion into account, we have adjusted the heading of Figure 8 in our document. This revision is reflected in lines 412-413 of the updated manuscript. Thank you for guiding us in improving our work.
“On”, delete “consistently”, “variables”, “maximum”
We apologize for our negligence. In our revised manuscript, the typo has been corrected in line 438, line 468, line 474 and line 475.
could you please clarify the meaning of this?
Thank you for your inquiry regarding the term "an artificial breeding base for Mangshan pit vipers." Allow me to clarify its meaning in the context of our study.
In China's "Fourteenth Five-Year Plan for Forestry and Grassland Conservation and Development," the Mangshan pit viper has been identified as a species that requires special conservation efforts. One of the key strategies proposed in this plan is the establishment of artificial breeding bases for this species. The goal of these bases is to increase the population of Mangshan pit vipers through controlled breeding, with the ultimate aim of preparing them for future reintroduction into the wild. We have added these contents in lines 79 to 93.
Our study aims to explore the optimal conditions for the breeding and rearing of Mangshan pit vipers. By conducting research on factors such as temperature, humidity, and other environmental conditions, we hope to provide data-driven support for the establishment and operation of large-scale artificial breeding programs. This, in turn, will inform the selection of suitable sites and the control of environmental conditions necessary for the successful breeding and rearing of this species.
We are grateful for your valuable feedback, which has helped us improve the clarity and accuracy of our manuscript. Thank you for your time and effort in reviewing our work. We believe that with these changes, the manuscript is now ready for publication.
Best regards,
